# Design Principles for Smart Linear Polymer Ligand Carriers with Efficient Transcellular Transport Capabilities

**DOI:** 10.3390/ijms25136826

**Published:** 2024-06-21

**Authors:** Ye Li, Zhun Zhang, Yezhuo Zhang, Jingcheng Hu, Yujie Fu

**Affiliations:** 1State Key Laboratory of Tree Genetics and Breeding, College of Biological Sciences and Technology, Beijing Forestry University, Beijing 100083, China; zhunzhang1202@bjfu.edu.cn (Z.Z.); zhangyezhuo@bjfu.edu.cn (Y.Z.); hujingcheng@bjfu.edu.cn (J.H.); 2Key Laboratory of Genetics and Breeding in Forest Trees and Ornamental Plants, Ministry of Education, College of Biological Sciences and Technology, Beijing Forestry University, Beijing 100083, China; 3The Tree and Ornamental Plant Breeding and Biotechnology Laboratory of National Forestry and Grassland Administration, College of Biological Sciences and Technology, Beijing Forestry University, Beijing 100083, China

**Keywords:** polymer, penetration, mechanism, delivery, computer simulation

## Abstract

The surface functionalization of polymer-mediated drug/gene delivery holds immense potential for disease therapy. However, the design principles underlying the surface functionalization of polymers remain elusive. In this study, we employed computer simulations to demonstrate how the stiffness, length, density, and distribution of polymer ligands influence their penetration ability across the cell membrane. Our simulations revealed that the stiffness of polymer ligands affects their ability to transport cargo across the membrane. Increasing the stiffness of polymer ligands can promote their delivery across the membrane, particularly for larger cargoes. Furthermore, appropriately increasing the length of polymer ligands can be more conducive to assisting cargo to enter the lower layer of the membrane. Additionally, the distribution of polymer ligands on the surface of the cargo also plays a crucial role in its transport. Specifically, the one-fourth mode and stripy mode distributions of polymer ligands exhibited higher penetration ability, assisting cargoes in penetrating the membrane. These findings provide biomimetic inspiration for designing high-efficiency functionalization polymer ligands for drug/gene delivery.

## 1. Introduction

In recent years, nanotechnology has rapidly progressed, leading to the widespread utilization of nanoparticles (NPs) in various applications such as phototherapy, cell imaging, and drug/gene delivery [1,2,3,4,5]. However, the cell membrane serves as a natural barrier to external substances, making the internalization mechanism of NPs into the cell membrane pivotal. To enhance drug/gene delivery efficacy, modifying functional polymer ligands is crucial to increase the targeted transport of drug molecules across the membrane and prolong their circulation in vivo [5,6]. Consequently, designing polymeric ligands that increase interaction with the cell membrane and improve gene/drug delivery is a noteworthy area of research.

In recent years, significant progress has been made in creating delivery systems using various biologically active agents, such as lipids [7], polymers [8,9,10], cell-penetrating peptides [11], nanoparticles (NPs) [12], and other biologically active agents [13,14,15]. Ligand shells can be added to the nanoparticle (NP) surface, thereby improving the adaptability and versatility of NPs in the cell environment, as required for a specific application [16,17,18,19,20]. Through experimental studies, the potential of ligand-tethered lipid NPs for targeted RNA delivery to activated fibroblasts has been demonstrated [21]. Hosoya et al. showed a broad range of ligand peptide-based applications with phage particles, heat-sensitive liposomes, or mesoporous silica NPs that self-assemble into a hydrogel aimed at tumor-targeted drug delivery [22].

Usually, nanomaterials used as drug/gene carriers enter the cell via endocytosis and penetration. In endocytosis, nanomaterials are engulfed within the vesicle, which is easy to target into the lysosome and hard to escape. However, polymers can usually directly penetrate the lipid membrane, which can escape from the lysosomes, avoiding degradation [23]. Due to their good design ability, biocompatibility, and high penetration ability, the use of different polymers as efficient nanocarriers for drug/gene delivery has attracted great attention recently [24,25]. These polymers are primarily composed of biocompatible and biodegradable polymers such as poly(lactic-co-glycolic acid) (PLGA), polyethylene glycol (PEG), and chitosan, among others [26,27,28,29]. A star-like polymer (glycidyl methacrylate) (s-PGMA)-based NP efficiently transferred loaded miRNA into heart tissues [21]. It has been found that poly(b-amino esters) (PBAEs), a type of polymer with lipid PEG, could functionally deliver mRNA to the lungs [30]. Low-molecular-weight polyethylenimine (PEI) is being developed to target the endothelial cells in the lung [31]. Studies have reported that cationic lipid-like material-coated NPs are capable of delivering DNA and siRNA to cells in culture [32]. It has been revealed that the uptake and translocation of gold NPs depend not only on the surface charge but also on the species of ligand present in the NPs [33]. Moreover, the distribution of ligands can also influence NP internalization [34,35]. By varying the ligand density and the surface charge of NPs, penetration, adsorption processes, and the final distribution of NPs can be easily modulated [34]. In previous work, we designed a cell-penetrating copolymer (CPC) carrier, which can definitely help the hydrophilic drug penetrate the lipid membrane effectively [36]. Although polymer-based NPs have fewer components, they offer greater structural flexibility [37]. The polymer ligand-modified NPs with tunable stiffness and identical surface chemistry can be selectively accepted by diseased cells based on their stiffness. Furthermore, NPs grafted with pH-responsive polymers can be accepted or rejected by cells due to the local pH environment [38]. Lipid molecules are another natural choice for adorning the NP surface, compared to polymer ligands. Lipid–polymer hybrid NPs (LNPs) have been reported to be another ideal drug carrier platform. The length of the hydrophobic anchor or aliphatic chain and the degree of saturation also influence transfection efficiency and cytotoxicity [39,40].

While previous experimental studies have demonstrated the ability of polymer ligands to enhance gene/drug delivery across cell membranes, there is a lack of systematic research on designing polymer ligands and their combination modes with high membrane penetration capabilities and low cytotoxicity [41]. Previous studies on the polymer ligand structure tend to overlook the fact that the type of polymer ligand is not only determinative in terms of targeting functionality, but factors such as length, stiffness, density, and distribution also play a significant role in delivery efficacy. The design of efficient polymers plays an important role in drug/gene transcellular transport, and with the rapid development of computational simulation, it provides a good prediction and design method. Moreover, the effective combination of computational prediction and experimental verification provides a more reliable technology for the design of efficient drug/gene carriers [42,43]. Thus, in this work, we applied dissipative particle dynamics (DPD) simulation to systematically study the properties of the polymer and its penetration ability across membranes. We investigated the effects of polymer stiffness, density, length, and combination mode on their penetration ability, which will provide a design strategy for the modification of polymer ligands for drug/gene delivery.

## 2. Results 

### 2.1. The Rigid Polymer Has a Higher Delivery Efficiency

To examine the influence of polymer stiffness on its penetration capability, different stiffness polymers were designed, namely K = 10 k_B_T, 30 k_B_T, 50 k_B_T, 80 k_B_T, and 100 k_B_T. In an experimental study, it was demonstrated that hydrophobic copolymers initially adhere to the lipid bilayer, and continued polymer surface penetration increases interfacial viscoelasticity [44]. Moreover, hydrophilic NPs are flattened in the membrane plane, while hydrophobic NPs are elongated along the membrane norm during penetration [45]. To balance water solubility and membrane penetration, both hydrophilic and hydrophobic polymers were constructed on the surface of the cargo. In varying systems, only the stiffness of the polymer was altered while keeping other parameters constant.

Initially, the polymers were attached on the cargo surface and placed on the membrane. Figure 1 displays the final snapshots of penetration for different systems after enough simulation time. Our simulations reveal that when the polymer stiffness is soft (K = 10 k_B_T, 30 k_B_T, and 50 k_B_T), and the hydrophobic polymer tends to hide within the hydrophilic polymer by curling in the solvent environment, which leads to a reduced penetration driving force. Ultimately, the cargo prefers to remain in the aqueous environment and is unable to penetrate the membrane (Figure 1A). However, when the polymer is more rigid (K = 80 k_B_T and 100 k_B_T), the polymers can assist the cargo in penetrating the lipid membrane and reaching the lower layer of the membrane (K = 80 k_B_T). The stiffness of the polymer significantly influences its penetration capability during cargo delivery. What causes this phenomenon? It is attributed to two primary factors: Firstly, increasing polymer stiffness restricts the polymer’s ability to adapt to the solvent environment, benefiting the hydrophobic polymer by providing the penetration driving force. Secondly, as the stiffness of the polymer ligand increases, the polymer is more likely to extend and interact with the lipid membrane, which promotes the driving force for the penetration of the polymer. The penetration process of the stiff polymers that assist the cargo across the lipid membrane is shown in Figure 1B. Monitoring the penetration process, we can see that initially, the hydrophilic polymer ligand interacts with the head region of the lipid membrane, followed by the insertion of the hydrophobic polymer into the tail region of the phospholipid membrane at 288 ns (Figure 1B). With the aid of both hydrophilic and hydrophobic polymers, the cargo penetrates the lipid membrane from the upper layer of the membrane to the lower layer via a “walkway” pathway. One should point out that, for stiff polymers during penetration, hydrophilic polymers grafted on the cargo surface can cause the surrounding phospholipids to become thinner, forming membrane pores (Figure 1B, t = 4960 ns). Figure 1C depicts the time evolution of the distance between the mass center of the cargo and the center of the bilayer, further confirming the finding that the penetration efficiency of the cargo is enhanced with rigid polymer ligands. Grafting the polymers on hydrophilic and hydrophobic cargo surfaces, it can be found that the rigid polymer can efficiently deliver both hydrophilic and hydrophobic cargo across the lipid membrane (K = 80 k_B_T and 100 k_B_T, Figure 1D). Moreover, previous studies have shown that in the presence of a particular ligand–receptor interaction, softer NPs undergo endocytosis at a much slower rate than their stiffer counterparts [46]. Thus, it is conceivable that softer NPs coupled with stiffer ligands might demonstrate superior efficacy in the intracellular delivery of cargo.

### 2.2. How to Design the Density of Polymers with Highly Efficient Transmembrane Ability

To examine the design principle of polymer density on the surface of cargo as drug delivery, we adjusted the ligand density from 0.3/nm^2^ to 2.1/nm^2^ by keeping the size of the cargo core constant and changing the number of modified ligands from 10 (*n* = 10) to 70. Our simulation results reveal that even with a lower polymer density (Figure 2A, *n* = 10), soft polymers can efficiently transport smaller-sized cargo (D = 3.2 nm) across phospholipid membranes. During the penetration process, the hydrophilic polymers interact with the head region of the lipid bilayer, followed by the gradual insertion of hydrophobic polymers into the hydrophobic core of the membrane. Finally, with the assistance of polymers, the smaller-sized cargo penetrates the lipid membrane (Figure 2B). Driven by entropy, the conformation of hydrophilic and hydrophobic polymers is spontaneously adjusted in the lipid membrane (Figure 2C). Moreover, increasing the polymer density results in faster transversion of the lipid membrane by the polymer-decorated cargo (Figure 2D). However, for medium-size cargo (D = 4.5 nm), when the density of soft polymers is 1.5/nm^2^ and the stiffness of polymer is 10.0 k_B_T, (Figure 1A), it fail to deliver medium-sized cargo across the lipid membrane. For sucressfully delivery of medium-sized cargo, it should increase the stiffness of the polymer. Furthermore, compared with hydrophobic cargo, hydrophilic cargo with the same size requires a higher polymer density for the transmembrane process (Figure 2E). 

### 2.3. How to Design the Length of Polymers with Highly Efficient Transmembrane Ability and Low Cytotoxicity

Polymer length is a crucial factor in designing drug/gene delivery carriers. In this work, we further investigated the impact of polymer length on their penetration ability. As shown in Figure 3, short polymers (1 bead, L = 0.6 nm) can only assist the hydrophobic cargo across the lipid membrane (Figure 3A, L = 0.6 nm). When the length of polymers is 2.6 nm (4 beads), it can carry the cargo across the membrane to the lower leaflet of the phospholipid membrane (Figure 3A, L = 2.6 nm). Increasing the polymer length enhances its ability to deliver cargo deeper into the lipid membrane. Moreover, with further increases in polymer length to six or eight beads (Figure 3A, L = 3.9 nm, 5.2 nm), a membrane pore is induced during the process of the polymer delivering the cargo to the lower leaf of the membrane. On the basis of the above results, we can infer that increasing the polymer length enhances its penetration depth and speed (Figure 3A,B). When the total length of the cargo and polymer is close to the thickness of the cell membrane, it promotes transmembrane transport. However, excessively long polymers may lead to the formation of membrane pores. These findings are consistent with our previous study indicating that the segment length of a transmembrane peptide polymer close to the lipid membrane thickness is optimal for copolymer penetration [36]. Compared to hydrophobic cargo, transporting hydrophilic cargo requires longer polymer assistance (Figure 3C).

Furthermore, we investigated the effect of cargo size on the translocation of the lipid membrane. Cargoes with sizes of 3.2 nm, 4.5 nm, 5.8 nm, and 7.1 nm were tested (Figure 4A). The simulation results show that only small cargo with a diameter of 3.2 nm can be carried across the membrane with a polymer length of 2.6 nm and a stiffness of 10 k_B_T. As the size of the cargo increased, the residence time in the solution environment was prolonged, resulting in a weakening of the ability for transmembrane transport (Figure 4A,B). To facilitate the transmembrane transport of larger cargoes, increasing the length and stiffness of the polymer ligand is needed.

### 2.4. How the Distribution of the Polymer Influences Its Delivery Ability across the Lipid Membrane

To examine the impact of hydrophilic and hydrophobic polymer distributions on their delivery capabilities, three types of polymer distributions were designed for the cargo (Figure 5A): (1) Janus/one-half distribution, where half of the polymers were hydrophilic, and the other half were hydrophobic; (2) one-fourth distribution; and (3) stripy distribution, where polymers alternated between hydrophilic and hydrophobic on the cargo surface, denoted as stripy 1, stripy 2, and stripy 3. The typical snapshots taken after sufficient simulation time are presented in Figure 5A. The one-half distribution polymers were unable to deliver the cargo into the lipid membrane. However, for the same number of hydrophilic and hydrophobic polymers arranged in a one-fourth distribution on the cargo surface, they were able to penetrate the lipid membrane spontaneously (Figure 5A). The details of the penetration process are illustrated in Figure 5B. First, the hydrophilic polymer on the surface of the cargo adheres to the lipid head, and then the hydrophobic polymer interacts with the hydrophobic regions of the cell membrane, driving the cargo into the cell membrane. During the penetration process, the hydrophilic polymer tends to curl up and hide (Figure 5B, 1984 ns). With the self-rotation, the polymers deliver the cargo, inserting deeply into the lipid membrane. Finally, to minimize the hydrophilic–hydrophobic mismatch, the hydrophobic polymer on the surface of the cargo is buried in the hydrophobic core of the membrane, while the hydrophilic polymer tends to extend to the position of the lipid head (Figure 5B, 2112 ns). For the stripy distribution condition, only polymers grafted on the cargo with the stripy 2 model can efficiently assist cargo penetration into the lipid membrane. This is because, compared with other stripy models, the stripy 2 model has more hydrophobic polymers, which provide a greater driving force. Our previous study has shown that stripy NPs with narrow stripes can gradually penetrate the cell membrane with less constrained rotation [47]. However, when polymer ligands are attached to the surface of the cargo, the system becomes more complex. On the one hand, polymers can adapt to the environment by themselves deforming, which can reduce the driving force to penetrate the lipid membrane, expecially for the hydrophilic and hydrophobic polymers are arranged on the cargo with narrow stripes. On the other hand, the rotation of the cargo becomes more difficult with the polymer modification on the cargo, which also can influence the penentration ability of the compound. 

To gain a better understanding of how polymer penetration ability is influenced by polymer properties, two-phase diagrams were created based on more than 27 independent simulations with varied polymer stiffness, lengths, and patterns (Figure 6). The results indicate that considering the hydrophilic cargo (D = 4.5 nm), when the polymer length is short (L = 0.6 nm), the hydrophilic cargo only adheres to the membrane (Figure 6A). Increasing the polymer length to 1.3 nm allows hydrophilic cargo to penetrate the lipid membrane regardless of polymer stiffness (Figure 6A). With a further increase in the polymer length (L = 2.6 nm, 3.9 nm, and 5.2 nm), the hydrophilic cargo prefers to be suspended in an aqueous solution when the polymer is soft (K = 10 k_B_T). Increasing the polymer stiffness can enhance the delivery capability of the longer polymers (Figure 6A). Additionally, the arrangement of polymers on the cargo can also influence their penetration ability. For the hydrophilic cargo with stiff polymers arranged in a one-fourth pattern and stripy 1 pattern, good penetration ability is observed (Figure 6B).

## 3. Discussion

In order to improve the ability of drug/gene to enter cells, some polymers are modified on the surface of drug/gene as carriers. At present, many types of polymers, such as dendritic polymers, cyclic polymers, linear polymers, starlike polymers are used as carriers, and the structure of the polymer also affects its fate into cell [48,49,50]. In experiment, it was found that for nonspecific clearance by the liver sinusoidal endothelium, when the liver sinusoids were treated with poly(ethylene glycol) (PEG)-conjugated oligo(l-lysine) (OligoLys), different polymer structures showed different results. OligoLys with a two-armed PEG configuration was ultimately cleared from sinusoidal walls to the bile, while OligoLys with linear PEG persisted in the sinusoidal walls [48]. For dendritic polymer, it is found that the low-generation dendrimer flattens on the gel-phase bilayer without any disturbance of lipids, while the high-generation dendrimers can induce the gel–fluid phase transition of lipids in the vicinity of the dendrimer and the significant bending of membranes [50]. Besides, Compared with shorter polymers, the longer polymers will deformation in the solution enviroments and during the process of the penetration. One should be noted that the change of configurational entropy for the polymer is also worth paying attention, which will make the transmembrane process more complicated. At same time, the charged nature of the polymer also affects its ability to enter cells. Usually, the surface of the cell membrane is negatively charged, some positively charged polymers, such as PEG and PEI, can promote the interaction with the cell membrane, and are benefited for the transport of cargo across cell membrane [51,52].

Except for the properties of the polymer, the environment in vivo is also quite complex, which has ion concentration gradients inside and outside the cell in vivo. Besides, when polymer modified cargo enter into the vivo due to their large surface areas and high surface reactivity, protein can spontaneously absorbed to the surface of polymer. These environmental factors can also affect the delivery efficiency of polymers [51,52,53]. Moreover, the cell membrane contains lipids and proteins in vivo.Weather the membrane proteins can influence the penentration of the polymer. All of these questions are worthy study in the future work. 

## 4. Materials and Methods

To simulate the transport ability of polymers, it is often difficult to achieve complete reproduction of real living cells due to the limitations of simulation. However, we captured the key factors of transmembrane transport and built a simplified model system that allowed us to effectively study the penetration ability of the polymer. In our simulation, coarse-grained models were used, which are presented in Figure 7. The cargo model was constructed by arranging dissipative particle dynamics beads in a spherical shape (denoted as C in Figure 7A) to represent the drug/gene, which was constrained to move as a rigid body. Both hydrophilic and hydrophobic cargoes were constructed. There are many types of line polymers, such as PEG, PEI, PCL, hyaluronic acid, and short peptides [54,55,56,57,58]. Usually, the polymer is modified on the surface of the cargo as a ligand to carry the cargo into the cell. In order to improve the solubility of cargo and its cycle time in the body, both hydrophilic and hydrophobic polymers were modified on the surface of the cargo. To better guide the selection of polymers as drug/gene carriers, we constructed coarse-grained models of polymers to study the properties of polymers and their ability to carry cargo across cells. The construction principle and process of its coarse-grained model are shown in Figure 7, which takes PEG as an example of a hydrophilic polymer ligand (Figure 7B–D). The coarse-grained polymer model (denoted as P in Figure 7D,E) was composed of a linear chain with multiple connected beads that were decorated on the surface of the cargo. Two types of polymers were examined: The hydrophilic polymer (denoted as P_H_ in Figure 7D) is shown in purple, and the hydrophobic polymer (denoted as P_T_ in Figure 7E) is shown in green. Polymer–cargo complexes were obtained by modifying polymer ligands on the surface of the cargo (Figure 7F). To systematically capture the properties of the polymer, we explored the effect of polymer stiffness (K), the length of the polymer (L), and the pattern of the polymer grafted on the surface of cargo on its penetration ability. All the parameters are variable. Unless specified, the hydrophobic-to-hydrophilic ratio of the polymer on cargo in this work is 1:1. For stripy 1 distribution, the hydrophobic-to-hydrophilic ratio of the polymer on cargo is about 1:2. For stripy 2 distribution, the hydrophobic-to-hydrophilic ratio of the polymer on cargo is about 2:1. Each amphiphilic lipid molecule (Figure 7G) was constructed by connecting a headgroup of three hydrophilic beads (denoted as H) to two hydrophobic tails of equal length, each possessing five hydrophobic beads (denoted as T) [59]. Although slightly distinct from the DMPC lipid model proposed by Grafmuller et al. [60,61], this model (Figure 7H) exhibits typical phase behaviors of lipid bilayers [62,63]. Solvent molecules were water molecules, which were coarse-grained into single beads (denoted as W) and explicitly included in the simulation system.

Dissipative particle dynamics (DPD) simulation was adopted in this study, which has been commonly used for studies of biomembrane systems [64,65,66,67,68,69], especially NP–membrane interactions [66,69]. In DPD, all elementary units are soft beads, which are governed by Newton’s equation of motion, d*r_i_*/d*t* = *v_i_* and d*v_i_*/d*t* = *f_i_*/*m_i_*, similar to the MD method. Typically, in DPD, there are three types of pairwise forces exerted on bead *i* by bead *j*, which are composed of a conservative force $F_{ij}^{C}$, a dissipative force $F_{ij}^{D}$, and a random force $F_{ij}^{R}$. Therefore, the total force exerted on bead *i* is expressed as follows:(1)$$F_{i}=\sum_{i\neq j}\left(F_{ij}^{C}+F_{ij}^{D}+F_{ij}^{R}\right)$$

The conservative force between beads *i* and *j* is determined as follows:(2)FijC=aijr^ijmax{1−rijrc,0}

In the simulation, the interaction parameter *a_ij_* (in $k_{B}T/r_{c}$) is the maximum repulsive force constant between beads *i* and *j*, ***r****_ij_* = ***r****_j_* − ***r****_i_* (***r****_i_* and ***r****_j_* are the positions of beads *i* and *j*), *r_ij_* = |***r****_ij_*|, ${\hat{r}}_{ij}$ = ***r****_ij_*/|***r****_ij_*|, and $r_{c}$ is the cutoff radius.

To study the penetration ability of polymer ligand-carried cargo, the interaction parameters between beads of the same type were set to $a_{WW}=a_{HH}=a_{TT}=25$, and those between the different types of beads were $a_{P_{H}H}=a_{P_{T}T}=a_{P_{H}W}=25$, $a_{P_{H}T}=a_{P_{T}H}=50$, and $a_{TW}=a_{P_{T}W}=80$. Note that in DPD, all interactions are soft and repulsive. If the interaction parameter is larger than 25 (the water–water interaction parameter), the interaction can be effectively regarded as repulsive. Otherwise, the interaction is attractive if the parameter is smaller than 25.

The dissipative force has the following form:(3)$$F_{ij}^{D}=-\gamma {\left(1-r_{ij}/r_{c}\right)}^{2}({\hat{r}}_{ij}\cdot v_{ij}){\hat{r}}_{ij}$$
where γ (in $\sqrt{mk_{B}T/r_{c}^{2}}$) is a simulation parameter related to the friction coefficient, and $v_{ij}=v_{i}-v_{j}$ ($v_{i}$ and $v_{j}$ are their velocities). This expression was chosen to conserve the momentum of each pair of particles and thus the total momentum of the system.

The random force also acts between each pair of beads as follows:(4)$$F_{ij}^{R}=\sigma (1-r_{ij}/r_{c})\theta_{ij}{\hat{r}}_{ij}$$
where *σ* represents the noise amplitude, and $\theta_{ij}$ is an uncorrelated random variable with zero mean and unit variance.

For each lipid, the interaction between neighboring beads within the same molecule is described by a harmonic spring force as follows:(5)$$F_{S}=K_{S}(r_{ij}-r_{eq}){\hat{r}}_{ij}$$
where the spring constant $K_{S}$ was set to 128, and the equilibrium bond length $r_{eq}$ was set to 0.7.

To maintain the bending rigidity of lipids, the force constraining the variation in bond angle is given by
(6)$$F_{\phi }=-\nabla U_{\phi }\;\mathrm{and}\;U_{\phi }=K_{\phi }((1-\cos\left(\phi {-\phi }_{0}\right))$$
where the bond angle ϕ is defined by the scalar product of two bonds connecting beads *i* − 1, *i*, *i* and *i* + 1. The equilibrium bond angle $\phi_{0}$ was set to π. $K_{\phi }$ = 10.0 is the bond bending constant [36].

All DPD simulations were performed with an ensemble of constant volume and temperature. Dimensionless units were used in this simulation. As usual, we chose the interaction cutoff radius $r_{c}$, the bead mass *m*, and the thermostat temperature *k_B_T* to be equal in the simulations. To map $r_{c}$ to its actual physical size, we used the formula proposed by Groot and Rabone [70], $r_{c}=3.107{\left(\rho N_{m}\right)}^{1/3}\;[Å]$. In this formula, $N_{m}$ is the number of water molecules represented by a DPD bead, and ρ is the density, i.e., the number of DPD beads per cube of volume $r_{c}^{3}$. As usual, it is assumed that $N_{m}$ = 3, ρ = 3, and a water molecule has approximately a volume of 30 ${Å}^{3}$. Hence, we determined $r_{c}$ to be 0.646 nm according to the formula. The system was evolved using a modified version of the velocity Verlet algorithm with a time step of Δt = 0.02. Periodic boundary conditions were implemented in all three directions. The simulation box size was 38.8 nm × 38.8 nm × 32.3 nm.

## 5. Conclusions

Through molecular simulations, this study uncovers design principles for polymers that efficiently penetrate membranes. We systematically investigated the impact of polymer stiffness, length, density, and arrangement on promoting cargo internalization. Our findings indicate that increasing the polymer ligand stiffness enhances cargo transport across membranes. However, excessively rigid polymers may cause a rupture of the lipid membrane. As the length of the polymer ligand increases, it can better facilitate cargo penetration into the lower layer of the lipid membrane. In particular, for the transportation of large cargo, the length and stiffness of the polymer need to be increased. Compared to hydrophobic cargo, hydrophilic cargo requires a higher polymer ligand density to enhance internalization. The polymer ligand distribution on the cargo surface also plays a crucial role in membrane transport. This study proposes a polymer design strategy for gene/drug carriers to enhance internalization capacity and reduce cytotoxicity.

## Figures and Tables

**Figure 1 ijms-25-06826-f001:**
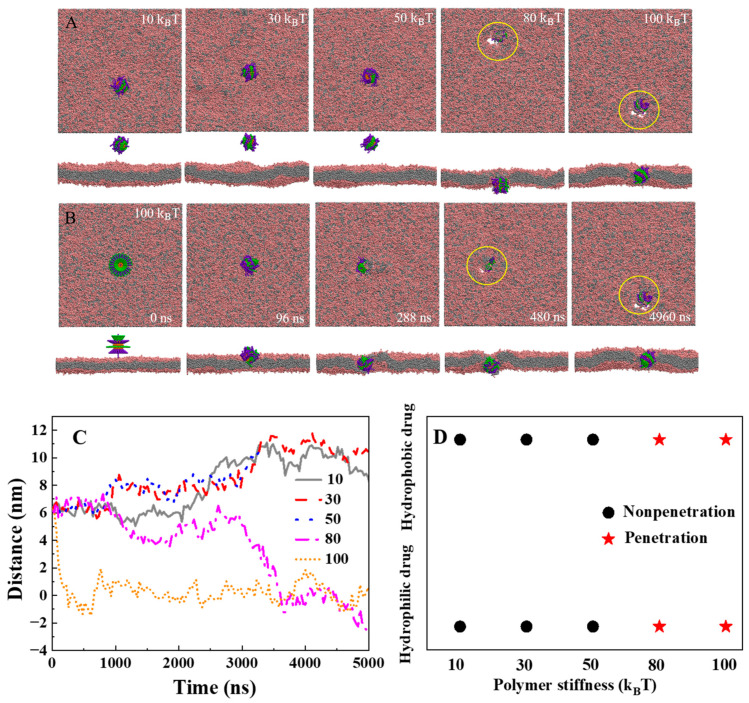
The effect of stiffness of polymer ligand on the cargo for the translocation of the lipid membrane: (**A**) The final typical snapshots for different stiffnesses of polymer ligand on the hydrophobic cargo. From left to right, the stiffnesses of polymer the ligand increase. K = 10 k_B_T, 30 k_B_T, 50 k_B_T, 80 k_B_T, and 100 k_B_T. In the snapshots, the lipid head is shown in pink, the lipid tail in gray, the hydrophobic cargo (D = 4.5 nm) in orange, the hydrophilic polymer in purple, the hydrophobic polymer in green, and the water molecule is not shown. The length of the polymer ligand is 2.6 nm. Locations of the membrane pores are labeled with the yellow circle. (**B**) Several typical snapshots during the penetration process for the hydrophobic cargo (K = 100 k_B_T). (**C**) The time evolution of the distance between the center of mass for the cargo and the center of the bilayer. (**D**) The internalization results for the polymers grafted on the hydrophilic cargo and hydrophobic cargo as the function of polymer stiffness.

**Figure 2 ijms-25-06826-f002:**
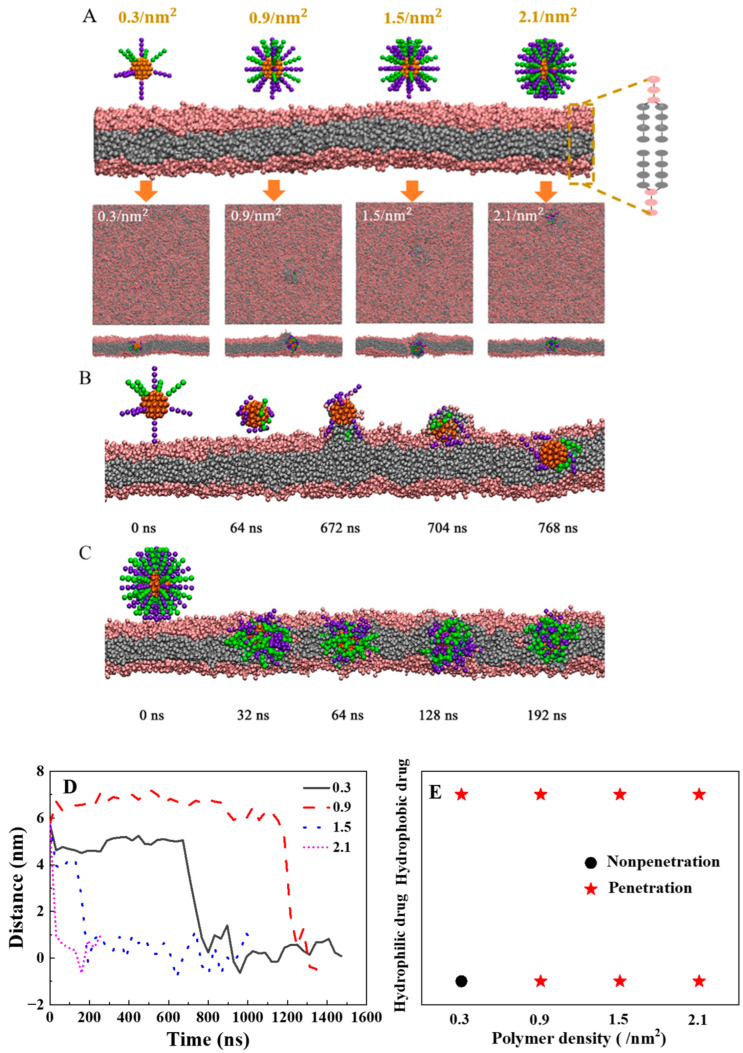
The effect of polymer ligand density on the cargo for the translocation of the lipid membrane: (**A**) The final typical snapshots for the penetration of different polymer ligand densities on the hydrophobic cargo. From left to right, the density of the ligand is 0.3/nm^2^, 0.9/nm^2^, 1.5/nm^2^, and 2.1/nm^2^. The diameter of the hydrophobic cargo is 3.2 nm. The stiffness of the polymer ligand is 10 k_B_T. The length of the polymer ligand is 2.6 nm. (**B**) Several typical snapshots during the penetrative process for the hydrophobic cargo with the polymer ligand density of 0.3/nm^2^. (**C**) Several typical snapshots during the penetrative process for the hydrophobic cargo with the ligand density of 2.1/nm^2^. (**D**) The time evolution of the distance between the center of mass for the cargo and the center of the bilayer. (**E**) The internalization results for the polymers grafted on the hydrophilic cargo and hydrophobic cargo as the function of polymer number.

**Figure 3 ijms-25-06826-f003:**
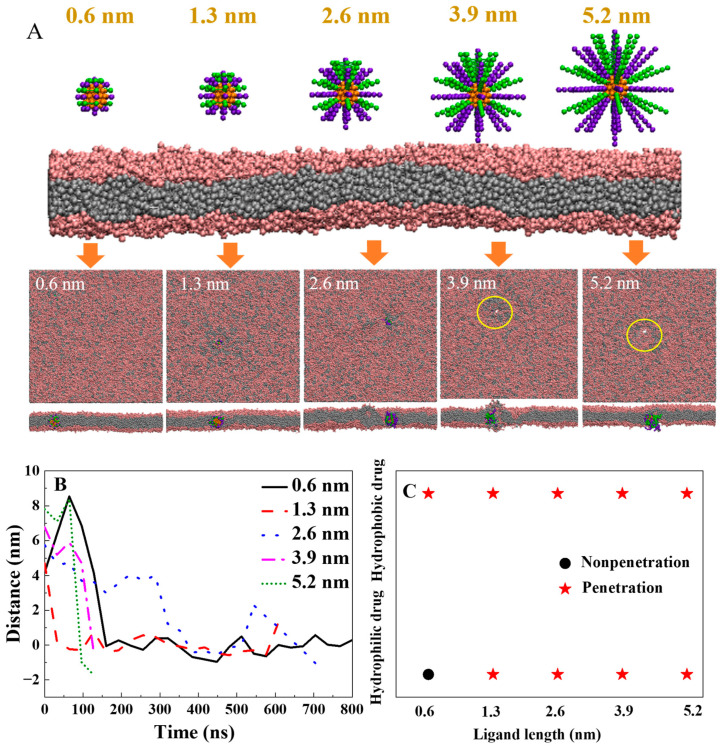
The effect of polymer ligand length on the hydrophobic cargo for translocation of the lipid membrane: (**A**) The final typical snapshot for the penetration of different polymer ligand densities of hydrophobic cargo. The lengths of the polymers from left to right are 0.6 nm, 1.3 nm, 2.6 nm, 3.9 nm, and 5.2 nm. The stiffness of the polymer ligand is 10 k_B_T. The diameter of the hydrophobic cargo is 3.2 nm. The ligand density is 1.5/nm^2^. Locations of the membrane pores are labeled with the yellow circle. (**B**) The time evolution of the distance between the center of mass for the cargo and the center of the bilayer. (**C**) The internalization results for the polymers grafted on the hydrophilic cargo and hydrophobic cargo as the function of polymer length.

**Figure 4 ijms-25-06826-f004:**
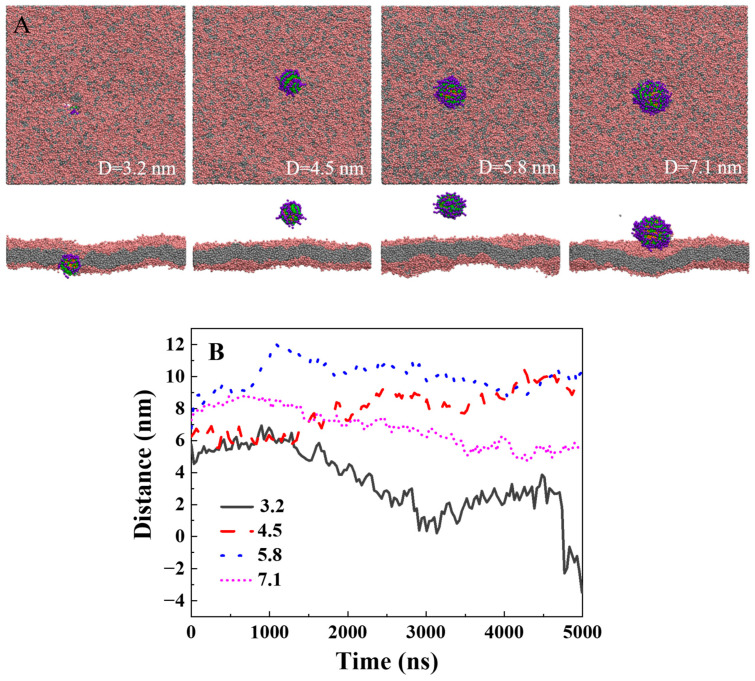
The effect of the size of the cargo on translocation of the lipid membrane: (**A**) The final typical snapshot for the penetration of cargoes with different diameters. The diameters of the hydrophobic cargoes from left to right are 3.2 nm, 4.5 nm, 5.8 nm, and 7.1 nm. The stiffness of the polymer ligand is 10 k_B_T. The polymer density is 1.5/nm^2^. The polymer ligand length is 2.6 nm. (**B**) The time evolution of the distance between the center of mass for the cargo and the center of the bilayer.

**Figure 5 ijms-25-06826-f005:**
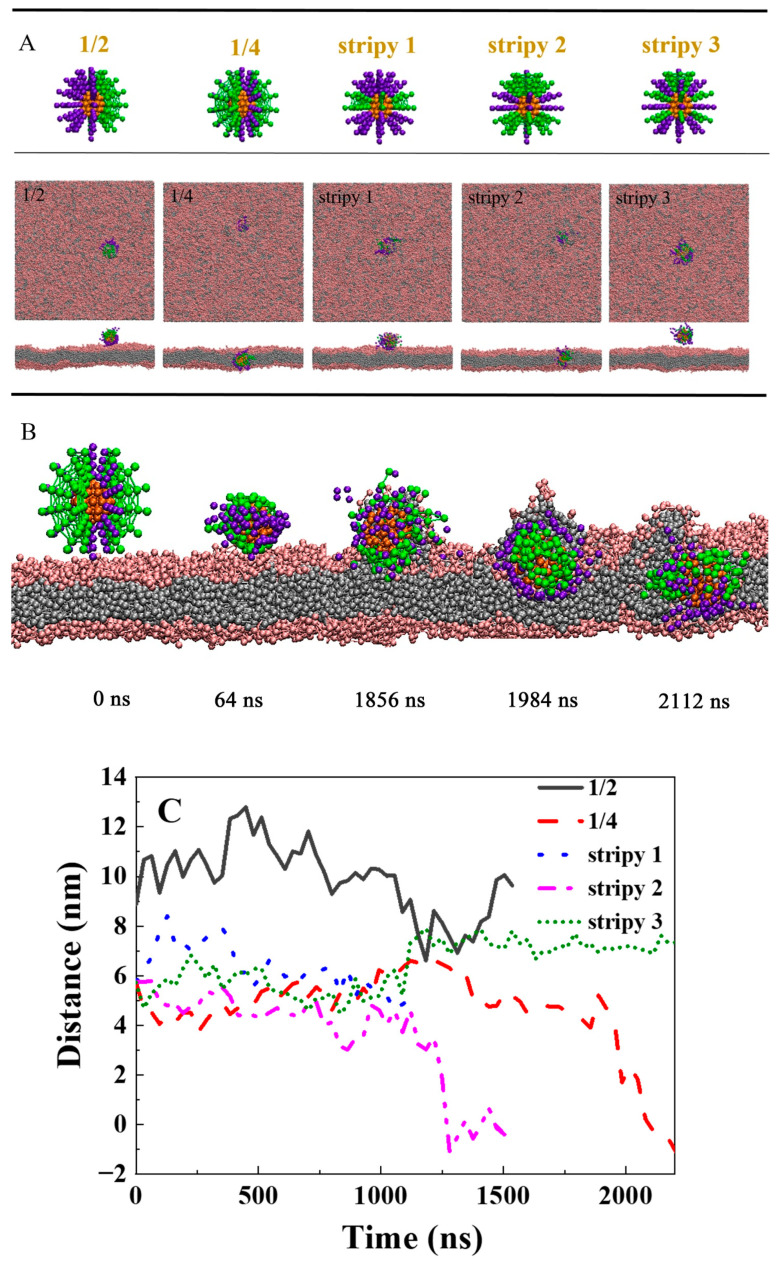
The effect of polymer ligand pattern on the cargo for the translocation of the lipid membrane: (**A**) The final typical snapshots of the penetration of different polymer ligand patterns on hydrophobic cargo. The diameter of the cargo is 3.2 nm. The stiffness of the polymer ligand is 50 k_B_T. The length of the polymer ligand is 2.6 nm. (**B**) Several typical snapshots during the penetrative process for the hydrophobic cargo with the polymer ligand with a 1/4 distribution pattern. (**C**) The time evolution of the distance between the center of mass for the cargo and the center of the bilayer.

**Figure 6 ijms-25-06826-f006:**
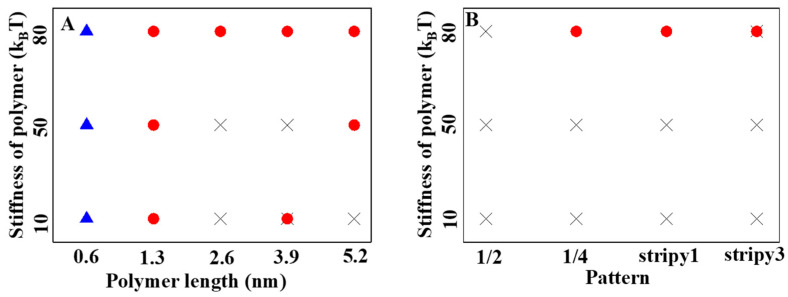
Phase diagrams summarizing the effect of polymer stiffness, polymer length, and polymer pattern on the internalization of hydrophilic cargo: (**A**) the effect of polymer stiffness and polymer length; (**B**) the effect of polymer stiffness and polymer pattern; ▲ represents the adhesion; ● represents the penetration; × represents the suspension.

**Figure 7 ijms-25-06826-f007:**
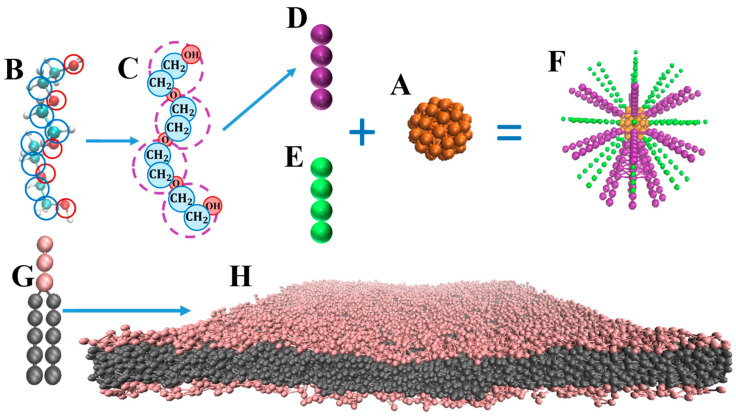
Schematic illustration of the coarse-grained models in the simulations: (**A**) cargo model; (**B**) hydrophobic polymer of all-atom model; (**C**) coarse-grained model of 13 agents; the groups in the purple dotted circles in the Figure 1C are coarse-grained into the purple beads, shown in Figure 1D; (**D**) coarse-grained model of 4 agents; (**E**) hydrophilic polymer model; (**F**) polymer–cargo complex model; (**G**) amphiphilic lipid molecule model; (**H**) lipid model.

## Data Availability

The data presented in this study are available on request from the corresponding author. The data are not publicly available due to [The amount of the datas is too many].

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
