# Peer review of "Design Principles for Smart Linear Polymer Ligand Carriers with Efficient Transcellular Transport Capabilities"

_ijms, 2024, doi:10.3390/ijms25136826_

Round 1
Reviewer 1 Report
Comments and Suggestions for Authors
This manuscript has a misleading title, there is no information about the polymers, any drug loading/encapsulation information. Furthermore, this has no chemistry/biology/molecular science related information and is unsuitable for this journal. IT is mostly a theoretical and computational study and provides no insights or novel information to this subject.
Most figures have either Y or X axis units missing (e.g. Figure 3 C, the legend mentions polymer length as beads, but polymer length is usually the number of repeating units. The same is seen for figure 2 E -density unit is missing). The quality of figures is also poor.
There is no information about what sort of polymers is being referred to-linear, dendritic, amiphilic etc.
Comments on the Quality of English LanguageModerate check needed. Sentences sometimes are broken up into may smaller ones and make no sense. writing should be improved.
Reviewer 2 Report
Comments and Suggestions for Authors
Y. Li and the team have undertaken a significant research endeavor, presenting a topic of utmost importance: “Design principles for smart and highly efficient polymers as drug/gene delivery carriers.” The manuscript delves into the crucial design principles underlying the surface functionalization of polymers using computational simulations. The team's focus on the stiffness, length, density, and distribution of polymer ligands that influence their penetration ability across cell membranes is a commendable contribution to the field. The manuscript holds promise for possible publication in MDPI’s IJMS. However, we request the authors to consider addressing the following issues and suggestions further to enhance the manuscript's potential impact across different disciplines.
1. The polymer (chemical structure, charge density, hydrophobic to hydrophilic ratio, etc.) used for computer simulations must be clearly described. 2. The authors should discuss the potential limitations of their computer simulation studies. 3. The authors should also discuss the discrepancy and limitations between in vitro and in vivo behavior of translocation across the membranes. 4. The manuscript focused on all parameters related to the membrane, including membrane adhesion, penetration, and translocation across the lipid membrane (transcytosis). However, the authors did not focus on disrupting endosomal and lysosomal membranes. Can these design principles be applied to endosomal and lysosomal membranes? Lines 37-38: We recommend the authors introduce the endosomal or lysosomal escape function of polymers. Endosomal and lysosomal escape, that is, translocating drug and gene therapeutic-carrying polymers from endosomes and lysosomes to the cytoplasm, is one of the significant hurdles of nanocarriers to efficient delivery. Hence, we recommend a brief discussion on endosomal disrupting polymers using the suggested citation (Macromol Rapid Commun 2022;43(12):e2100754). 5. A recent focus on addressing the gap between experiments and simulations. This article will help the authors to address the gap between experiment and simulation studies of polymeric carriers (Mol Pharm. 2024. doi: 10.1021/acs.molpharmaceut.3c00747).
6. The membrane translocation also depends on the configuration. For example, the configuration of PEG (linear one-arm versus branched two-arm), which is attached to an oligopeptide, influenced the translocation and clearance behavior of PEG polymers across the lipid bilayers of liver sinusoidal endothelial cells. Both one- and two-arm-PEG-oligopeptides attached to liver sinusoidal wall cells; however, oligopeptide having a branched two-armed PEG configuration was ultimately translocated and cleared from sinusoidal walls to the bile, whereas oligopeptide with linear one-armed PEG persisted in the sinusoidal walls. This information would fit the introduction part. Please discuss. This study (Sci Adv 2020;6(26):eabb8133) provides a relevant example of how polymer configuration can influence translocation, supporting the need for a discussion on this topic in the manuscript. 7. Parameters such as temperature, ionic strength, salt concentration, and volume must be clearly mentioned with units in DPD simulation studies.
Round 2
Reviewer 1 Report
Comments and Suggestions for Authors
The authors did not address major concerns in the previous review round
Comments on the Quality of English LanguageSpell check and grammar check needed
Reviewer 2 Report
Comments and Suggestions for Authors
Page 6, Lines 199 - 203: Reference should be corrected in the manuscript. Add the correct reference for the sentence "The structure of the polymer also affects its fate".
In response cover letter, the authors used the correct reference. However, the authors failed to add the correct reference in the original manuscript. Please add the correct reference.
Round 3
Reviewer 1 Report
Comments and Suggestions for Authors
The title of the manuscript is misleading and must be changed as this is not a chemistry or polymer manuscript and has no chemical synthesis, characterization, related studies. The authors argue that polymers are chemical substances but this is not logical enough to use it in the main title as if a new polymer has been made or a class of polymers is being studied chemically/biologically.
The figure themselves have not been corrected (for example the X and Y axes in figure 3 and 7 lack units)
Comments on the Quality of English LanguageMinor revision typo corrections needed
Round 4
Reviewer 1 Report
Comments and Suggestions for Authors
Revisions made as per the capacity of the authors
Comments on the Quality of English LanguageMinor spell check/typo/grammar editing needed
